# Mechanism of Caspase-1 Inhibition by Four Anti-inflammatory Drugs Used in COVID-19 Treatment

**DOI:** 10.3390/ijms23031849

**Published:** 2022-02-06

**Authors:** Francesco Caruso, Jens Z. Pedersen, Sandra Incerpi, Sarjit Kaur, Stuart Belli, Radu-Mihai Florea, Miriam Rossi

**Affiliations:** 1Department of Chemistry, Vassar College, Poughkeepsie, NY 12604, USA; sakaur@vassar.edu (S.K.); belli@vassar.edu (S.B.); rflorea@vassar.edu (R.-M.F.); rossi@vassar.edu (M.R.); 2Department of Biology, University Tor Vergata, 00133 Rome, Italy; j.z.pedersen@gmail.com; 3Department of Sciences, University Roma Tre, 00146 Rome, Italy; sandra.incerpi@uniroma3.it

**Keywords:** COVID-19, caspase-1, cytokine storm, anti-inflammatory drugs

## Abstract

The inflammatory protease caspase-1 is associated with the release of cytokines. An excessive number of cytokines (a “cytokine storm”) is a dangerous consequence of COVID-19 infection and has been indicated as being among the causes of death by COVID-19. The anti-inflammatory drug colchicine (which is reported in the literature to be a caspase-1 inhibitor) and the corticosteroid drugs, dexamethasone and methylprednisolone, are among the most effective active compounds for COVID-19 treatment. The SERM raloxifene has also been used as a repurposed drug in COVID-19 therapy. In this study, inhibition of caspase-1 by these four compounds was analyzed using computational methods. Our aim was to see if the inhibition of caspase-1, an important biomolecule in the inflammatory response that triggers cytokine release, could shed light on how these drugs help to alleviate excessive cytokine production. We also measured the antioxidant activities of dexamethasone and colchicine when scavenging the superoxide radical using cyclic voltammetry methods. The experimental findings are associated with caspase-1 active site affinity towards these compounds. In evaluating our computational and experimental results, we here formulate a mechanism for caspase-1 inhibition by these drugs, which involves the active site amino acid Cys285 residue and is mediated by a transfer of protons, involving His237 and Ser339. It is proposed that the molecular moiety targeted by all of these drugs is a carbonyl group which establishes a S(Cys285)–C(carbonyl) covalent bond.

## 1. Introduction

The mechanism of COVID-19 development remains unclear, although unregulated cytokine release resulting in an excessive inflammatory response is likely implicated. Several drugs previously used to modulate immune responses in many diseases, including two readily available and inexpensive corticosteroids, dexamethasone and methylprednisolone, have now been repurposed and used as promising new tools for reducing the inflammatory response to COVID-19 virus infection [1,2]. “Cytokine storm” and cytokine release syndrome are life-threatening systemic inflammatory syndromes involving elevated levels of circulating cytokines and immune-cell hyperactivation that can be triggered by various therapies, pathogens, cancers, autoimmune conditions and monogenic disorders [3]. These damaging conditions prompt the formation of inflammasomes, large multi-protein complexes that include the protease caspase-1. Caspase-1 cleaves interleukins, pro-IL-1β and pro-ΙL-18, into their operative cytokine forms, and also cleaves the protein gasdermin D, which triggers pyroptosis, a pro-inflammatory form of cell death. These are likely implicated in acute COVID-19 pathogenesis and may be more damaging to the host than the response associated with viral-induced cell death in the host [4]. An intensified immune response may be a major contributor to the poor outcomes often observed in patients with COVID-19 [5]. Figure 1, taken from [6], shows the mechanism of inflammasome activation.

Activation of the inflammasome is a key function mediated by the innate immune system [7]. Several families of PRRs are important components in the inflammasome complex, including the nucleotide-binding domain, leucine-rich repeat containing proteins (NLRs, also known as NOD-like receptors). Upon sensing certain stimuli, the relevant NLR can oligomerize to be a caspase-1-activating scaffold. Active caspase-1 subsequently functions to cleave the proinflammatory IL-1 family of cytokines into their bioactive forms, IL-1β and IL-18, and cause pyroptosis, a type of inflammatory cell death [8,9].

In this study, we suggest that inhibition of caspase-1, an enzyme involved in triggering the release of pro-inflammatory cytokines, is relevant to the mechanism of anti-COVID-19 therapy [6]. Since there is a strong suspicion that excessive amounts of cytokines create damage due to the extreme inflammatory responses they produce, such as high levels of the pro-inflammatory cytokine IL-1β in the lung parenchyma [10] and increased levels of IL-18 in sepsis-induced acute respiratory distress syndrome (ARDS) [11], we propose that inhibiting the caspase-1 protease might decrease the cytokine storm and ameliorate patient healing. Figure 2 shows the potential inhibitors of caspase-1 in this study.

Colchicine, an alkaloid isolated from *Colchicum autumnale*, is among the oldest plant drug compounds known and has been used for centuries to treat gout illness. It inhibits polymerization of microtubules [12]; earlier studies showed that different concentrations of colchicine have diverse effects on the stabilization, dynamics and inhibition of microtubule polymerization [13,14,15]. At low doses (about 10^−8^ M), the effects are associated with gout inflammation and treatment, and the direct interaction of colchicine with tubulin monomers, or with whole microtubules, leads to protein conformational changes and destabilization. At high doses (about 10^−6^ M), a second mechanism involves suppression of enzymes that may lead to inflammatory restraint, including caspase-1. Since caspase-1 cleaves pro-IL-1, the release of mature IL-1 decreases upon administration of colchicine, thus inhibiting the propagation of inflammation [16]. Colchicine is used as a treatment for rheumatoid arthritis, a pro-inflammatory cytokine-driven chronic articular condition often accompanied by cardiovascular and lung pathologies. Intriguingly, colchicine shows activity against COVID-19 illness [17], resulting in inhibition of the viral activation of the inflammasome. SARS-CoV-2 drives the assembly of a pro-inflammatory protein complex, the nod-like receptor protein 3 (NLRP3) inflammasome [18], which is composed of the sensor NLRP3, the apoptosis-associated speck-like protein adaptor (ASC or PYCARD) and the effector caspase-1 [19]. Caspase-1 converts pro-interleukin (IL)-1β and pro-IL-18 to their active forms and activates gasdermin-D, permitting large-scale secretion of IL-1β with a subsequent increase in large quantities of pro-inflammatory cytokines [20,21], IL-1β, tumor necrosis factor (TNF) and ligation of Toll-like receptors that activate NF-κB [18] to further upregulate the inflammasome. In SARS-CoV-1, a small envelope protein (E) enhances this reaction that promotes further assembly and activation of the NLRP3 inflammasome [22]. In addition, the creation of IL-1β drives the synthesis of IL-6, a cytokine that induces C-reactive protein (CRP) and which has been identified as a major pro-inflammatory agent in the COVID-19 cytokine storm [23,24,25,26]. Colchicine reduces the concentration of inflammatory biomarkers observed in moderate-to-severe COVID-19 patients and is considered a strong candidate for adjunctive COVID-19 treatment [27]. In eight studies involving 926 COVID-19 patients, 520 patients had standard-of-care therapy while 406 patients also received colchicine and the mortality rate was significantly lower in the colchicine group compared to the control group [28]. Thus, exploring the potential inhibition of caspase-1 by colchicine provides useful information to ameliorate the health condition of COVID-19 patients.

Dexamethasone is a cheap and globally available corticosteroid commonly used as an anti-inflammatory and immunosuppressant. It may modulate inflammation-mediated lung injury and thereby reduce progression to respiratory failure and death. One study showed that, in COVID-19 patients, administration of dexamethasone reduced deaths by one third in patients requiring a ventilator, and one fifth in other patients receiving oxygen only [29]. Dexamethasone also significantly inhibited the activity of the NLRP3 inflammasome and reduced the protein contents of pro-caspase-1, caspase-1 and caspase-1/pro-caspase-1, as well as the cytokines IL-1β, IL-6 and IL-17 in lung tissues. [30]

Since the corticosteroids dexamethasone and methylprednisolone assist in the recovery of severely infected COVID-19 patients needing respiratory support [31], it is of interest to analyze whether the inhibition of caspase-1 is implicated in these cases. Dexamethasone was reported to inhibit LPS-induced inflammation and lung damage by inhibition of the NLRP3 inflammasome and therefore caspase-1 and inflammatory cytokines in mouse macrophages [32]. The subtle structural differences among the three corticosteroids used in COVID-19 treatments (dexamethasone, methylprednisolone and hydrocortisone) compared to inactive anti-inflammatory drugs, have been discussed recently by Draghici and Mor [33]. Comparisons between dexamethasone and methylprednisolone have been carried out by several scholars [34,35,36]. In their paper, Draghici et al. found methylprednisolone more effective than dexamethasone and hydrocortisone, as predicted by in silico evaluation [33]. Other papers have shown the greater effectiveness of methylprednisolone with respect to dexamethasone. In a recent review, an extensive description of clinical trials was discussed, and glucocorticoid action was suggested regarding immunomodulation as a potent COVID-19 pharmacotherapy [37]. As to hydrocortisone, the data reported so far are not clear [38] and will not be further evaluated in the present paper.

The selective estrogen receptor modulator (SERM), raloxifene, has been in use since 1997 for treatment of postmenopausal osteoporosis and breast cancer in the US. Existing clinical studies indicate that estrogens and SERMs, such as raloxifene, modulate immune responses. In particular, raloxifene demonstrates immunoprotective effects [39]. A recent review emphasizes that raloxifene also shows anti-viral activity towards RNA viruses, such as Ebola, influenza A and hepatitis C viruses [40]. Allegretti et al. have performed in vitro experiments that confirm raloxifene to be the most active SERM against SARS-CoV-2 virus at low (micromolar) concentrations [41]. As a result, raloxifene may be usefully repurposed as a drug against COVID-19 [41,42,43]. The in vivo anti-inflammatory effects of raloxifene are described in a study on ovarietomized mice, in which its administration decreased proteinuria-induced renal tubular damage [44]. These mice showed activation of tubular inflammasomes and increased levels of caspase-1-mediated cytokines IL-1β and IL-18. Raloxifene administration resulted in diminished concentrations of these pro-inflammatory cytokines and in less cellular damage.

The chemical interactions among the main COVID-19 protease and the many anti-COVID-19 molecules show the essential role of the active site cysteine amino acid as it interacts with a carbonyl moiety of the inhibitor, such as in ketoamides [45], quinones [46] and quinone methides [47]. Our interest in describing the inhibition of caspase-1, also a cysteine protease, through interaction with an inhibitor carbonyl moiety arises from this relationship. In this study, all four medicinal compounds, two drugs used against COVID-19, dexamethasone and methylprednisolone, plus colchicine and raloxifene, two other medicinal compounds showing anti-COVID-19 activity, contain potentially reactive carbonyl groups, and so form an interesting set to explore the mechanisms by which these drugs act against COVID-19. For all four drugs, our computational results from molecular mechanics, DFT plus HOMO and LUMO calculations, revealed that caspase-1 formed covalent bonds with the critical cysteine residue at the enzyme active site.

In addition, a recent study by us showed that the formation of a covalent bond between a SARS-CoV-2 inhibitor, celastrol and the Cys145 thiolate at the main COVID-19 protease active site, can be associated with the antioxidant activity of celastrol when scavenging the superoxide radical [47]. Therefore, experimental measurement of dexamethasone and colchicine antioxidant activity towards the damaging superoxide radical using the RRDE cyclovoltammetry method is also described. From these results, we found the activation of a protonated carbonyl, which supports the molecular mechanism described in Scheme 1.

## 2. Results and Discussion

### 2.1. Docking Studies

#### 2.1.1. Dexamethasone

Docking of dexamethasone on caspase-1 was performed using the PDB 6PZP receptor [48], which consists of two protein subunits containing the inhibitor VX-765. After application of a CHARMm forcefield [49] in Discovery Studio, VX-765 was selected to create a sphere of radius 10 Å. The inhibitor was then eliminated and dexamethasone docked to obtain 15 poses. To perform this docking calculation, the crystal structure atomic coordinates of dexamethasone were used as taken from the CSD Database (refcode DEXMET11) [50].

Examination of the results after docking of dexamethasone at the active site of the caspase-1 receptor reveals some important interactions with Cys285 in the structurally closely related poses 2, 13 and 14. These include a S(Cys285)–C(carbonyl) distance of 3.202 Å (pose 13) (Figure 3 and Figure 4). In addition, the H-bond between F(dexamethasone) and H_2_N(Arg-B-341), 2.745 Å, provides stabilization of this cysteine–dexamethasone complex, which is absent in pose 2 and pose 14.

The Discovery Studio molecular mechanics program provides the CDocker interaction energy, which is the non-bonded interaction energy (composed of the van der Waals term and the electrostatic term) between the protein and the ligand related to the force field CHARMm. This parameter is −23.9 kcal/mol for dexamethasone pose 13. Figure 5, Figure 6 and Appendix A show HOMO and LUMO of dexamethasone.

#### 2.1.2. Methylprednisolone

Methylprednisolone is another corticosteroid with anti-inflammatory activity in use for COVID-19 patients [51] and is structurally related to dexamethasone (Figure 2). We used crystal structure atomic coordinates from the CSD database (refcode MTHPRG) [52] for docking methylprednisolone. Pose 8 of this anti-inflammatory drug shows the interaction between S(Cys285) and C(carbonyl) of methylprednisolone equal to 3.135 Å. Further structural details are shown in Figure 7 and Appendix A. CDocker interaction energy is −24.0 kcal/mol for pose 8.

#### 2.1.3. Colchicine

We next explored colchicine using crystal structure atomic coordinates from the CSD database (refcode COLCDH) for docking [53]. Pose 9 shows the interaction between S(Cys285) and the tropolone C(carbonyl) of colchicine, 3.363 Å, and its CDocker interaction energy is −31.2 kcal/mol (Figure 8 and Figure 9). CDocker interaction energy of pose 1 is −38.5 kcal/mol. Figure 10, Figure 11 and Appendix A provide HOMO and LUMO of colchicine.

#### 2.1.4. Raloxifene

Since our docking results repeatedly showed interaction between S(Cys285) and a C(carbonyl), we further enlarged our analysis by investigating also raloxifene, a medicinal compound having anti-inflammatory activity in clinical use against COVID-19 infections which contains a carbonyl functional group. Raloxifene atomic coordinates were taken from CSD (refcode SAQYIR) [54]. Docking of raloxifene into the caspase-1 enzyme (Figure 12 and Appendix A) shows pose 2 having an interaction between S(Cys285) and C(carbonyl) of raloxifene of 3.833 Å, whose CDOCKER interaction energy is −39.1 kcal/mol.

### 2.2. Caspase-1 Inhibitory Mechanism

Overall, these medicinal compounds having anti-inflammatory activity are very promising for treating symptoms of COVID-19 disease, but their chemical mechanism has not yet been clarified. In this study, we take advantage of indications in the literature that inhibition of caspase-1, an enzyme involved in triggering the release of cytokines, may be relevant to the mechanism of anti-COVID-19 therapy. That is, inhibiting caspase-1 may decrease the storm of cytokines involved in response to the SARS-CoV-2 virus attack. The NLRP3 inflammasome promotes inflammation via caspase-1-mediated cleavage and activation of key inflammatory molecules, including active caspase-1, IL-1β and IL-18. It has been demonstrated that the NLRP3 inflammasome is activated in response to SARS-CoV-2 infection and is active in COVID-19 patients [55]. In post-mortem tissues of moderate and severe COVID-19 patients, active NLRP3 inflammasomes have been found upon autopsy. Inflammasome-derived products, such as caspase-1 and IL-18, in sera correlates with the markers of COVID-19 severity, including IL-6. Moreover, higher levels of IL-18 and caspase-1 are associated with disease severity and poor clinical outcome [55]. Since there is strong evidence that production of excessive cytokines creates health problems in COVID-19 patients [55], inhibiting the caspase-1 protease may help patient healing. In fact, the cleaved (active form) of caspase-1 exerts its catalytic activity on the pro-inflammatory cytokines that, after their release, perpetuate the inflammatory response [56]. The docking of four drugs (Figure 1) into the caspase-1 active site highlights promising features and directs us to propose the mechanism shown in Figure 1 below.

Formation of the S(Cys285)–C(carbonyl) covalent bond is the key feature of this mechanism, as described in three steps:(1)HN(His237) proton capture by the medicinal substrate O(carbonyl). This is suggested by a strong H-bond between His237 and O(carbonyl) in all four substrates, with the following distances: 2.165 Å (dexamethasone), 1.926 Å (methylprednisolone), 1.956 Å (colchicine) and 2.484 Å (raloxifene).(2)H-bond distance of 2.818 Å in the 4 docking results suggest proton transfer from H(Cys285) to O(Ser339), creating S(thiolate) through Cys285 S–H cleavage. Indeed, a small variation in the cysteine C-C-S-H torsion angle from the docked −60°, corresponding to 2.818 Å, to −20°, makes this H-bond shorter, 2.582 Å, further suggesting proton transfer.(3)The S(thiolate) directs its attack on the positively charged (C=OH)**^+^** moiety of these anti-inflammatory molecules (step 1), suggesting the formation of a S–C covalent bond. This is supported by the following distances to S(Cys285): 3.202 Å (dexamethasone), 3.135 Å (methylprednisolone), 3.363 Å (colchicine), 3.833 Å (raloxifene). A consistent molecular mechanism is thus designated using computational methods for dexamethasone, methylprednisolone, colchicine (currently used in treatment for COVID-19) as well as raloxifene, which, from the results of this study, are predicted to be promising clinical candidates for use in COVID-19 cases.

Dexamethasone possesses one carbonyl group on the quinone methide moiety and another at the other end of the molecule. HOMO and LUMO (Appendix A, Figure 5 and Figure 6) suggest that the former carbonyl is more prone to reactivity in a S-thiolate nucleophilic attack. It is worth noting that the stabilizing role of F(dexamethasone) (Figure 3 and Figure 4 may indicate an important difference with methylprednisolone and potentially explain the superior performance of dexamethasone against COVID-19.

After docking the four medicinal compounds into the caspase-1 receptor, the aforementioned mechanism (steps 1–3) illustrates some important differences, however. One is seen in the S–C(carbonyl) interaction for raloxifene, 3.833 Å, which is markedly longer and thus weaker, than for the other three compounds. However, the CDocker interaction energy, −39.1 kcal/mol, is the most negative of all these four compounds, and this positive feature is due to many important interactions (Appendix A). This may indicate the stability of the raloxifene–caspase-1 complex, even though the effective S(Cys285)–caspase-1 covalent bond may be difficult to form. Interestingly, a recent investigation shows that raloxifene inhibits IL-6 signaling at therapeutic doses, suggesting the potential to prevent the COVID-19 cytokine storm [43]. In addition, the use of raloxifene in clinical trials for patients with mild COVID-19 symptoms has been authorized by the Italian pharmaceutical regulatory agency, AIFA [57]. We conclude that the formation of a covalent bond between the S(Cys285) and C(carbonyl) of raloxifene remains an open question and requires further study. We calculated HOMO and LUMO for colchicine (Figure 10 and Appendix A). Both figures show that the acetamide carbonyl is not involved.

### 2.3. DFT and RRDE

The main COVID-19 protease contains an active site cysteine group prone to establish a thiolate nucleophilic attack on the O(carbonyl) belonging to the protease inhibitor celastrol [47]. This action can be associated with the superoxide anion (mimicking S(thiolate)) scavenging by the celastrol quinone methide ring when positively charged. In this study we envision a nucleophilic approach of S(Cys285)-thiolate to the carbonyls belonging to the four medicinal agents and so we explore the related superoxide scavenging. Figure 13 shows DFT relevant bond distances at the quinone methide ring of dexamethasone. After the superoxide approaches the quinone methide ring (Figure 14), it moves away and does not release its electron to the ring. However, upon dexamethasone carbonyl protonation (Figure 15), the superoxide is able to release its electron, generating molecular O_2_, which is eliminated from dexamethasone (Figure 16). A further DFT analysis shows that a protonated dexamethasone is sensitive to superoxide (Appendix A), as formation of O_2_H is seen after geometry optimization for both components at initial van der Waals σ separation, i.e., not involving π–π interaction.

RRDE analysis of dexamethasone (Figure 17) shows that superoxide concentration does not decrease upon adding increasing amounts of dexamethasone, which indicates no scavenging. The slight decrease of superoxide, seen at the beginning of the curve (after adding the first 5 μL aliquot of dexamethasone), is assigned to the small amount of water in the DMSO solvent. We conclude that dexamethasone is not a good scavenger for superoxide, and protonated dexamethasone existence is suspected from the RRDE experiment.

In contrast with dexamethasone, colchicine is markedly prone towards capturing a proton, according to DFT calculations. Thus, Figure 18 and Figure 19 show that the system formed by the van der Waals π–π separated superoxide and colchicine evolves towards the shortening of O–O bond distance in the superoxide, plus important structural modifications in the tropolone ring: shortening of C–C bond distances and lengthening of C=C bond distances, due to capture of the superoxide electron. The arrival of a proton to the negatively charged tropolone ring stabilizes the hydroxyl formation of colchicine (Figure 20), which in turn has further reactivity towards the superoxide (Figure 21). Interestingly, a σ attack of superoxide on the system shown in Figure 20 also seems feasible (Appendix A). Therefore, the association of a thiolate nucleophilic attack on protonated colchicine at the active site of caspase-1, similar to that found for celastrol binding at the main protease of COVID-19 [36], may be expected. We plan to perform further cyclovoltammetry studies with the other drugs explored in this study.

Cyclovoltammograms of colchicine are shown in Figure 22 and the collection efficiency (Figure 23) shows that colchicine is a good scavenger of superoxide. The linear behavior seen in Figure 23 gives a slope of −3.3 × 10^4^, which defines colchicine as a weaker scavenger than quercetin, whose slope is −6.0 × 10^4^ [58]. It is interesting that the commercially used antioxidant BHT has a less steep slope than colchicine: −1.6 × 10^3^ [58]. Therefore, the correlation between a thiolate nucleophilic attack on a protonated carbonyl (at the caspase-1 active site) and the anion superoxide affinity for the protonated colchicine confirms the equivalent situation described recently for celastrol [47], an active natural product against COVID-19. Regarding dexamethasone, its null RRDE antioxidant action (Figure 17) does not preclude superoxide reactivity towards a protonated carbonyl of dexamethasone, as shown by the DFT study (Figure 15, Figure 16 and Appendix A) and observed at low concentration in Figure 17.

## 3. Materials and Methods

Dexamethasone and colchicine were from Sigma-Aldrich, St. Louis, MO, USA.

### 3.1. Equipment

Hydrodynamic voltammetry at a rotating ring-disk electrode (RRDE) was carried out using the Pine Research WaveDriver 20 bipotentiostat (Pine Research, Durham, NC, USA) with the Modulated Speed Electrode Rotator. The working electrode is the AFE6R2 gold disk and gold ring rotator tip (Pine Research, Durham, NC, USA) combined with a coiled platinum wire counter electrode and a reference electrode consisting of an AgCl coated silver wire immersed in 0.1 M TBAB in dry DMSO in a fritted glass tube. The electrodes were placed in a 5-neck electrochemical cell together with means for either bubbling or blanketing the solution with gas. Voltammograms were collected using Aftermath software provided by Pine Research. Careful cleaning of the electrodes was performed by polishing alumina particle suspension (Allied High Tech Products, Inc.; Rancho Dominguez, CA, USA) on a moistened polishing microcloth to eliminate potential film formation [60]. Further details of this technique, developed by our lab, are available [61].

### 3.2. Calculations

Calculations were performed using programs from Biovia (SanDiego, CA, USA). Docking studies were applied using the CDOCKER package in Discovery Studio 2020 version [49]. Density functional theory (DFT) code DMol3 was applied to calculate energy, geometry, HOMO, LUMO and frequencies implemented in Materials Studio 7.0 [62]. We employed the double numerical polarized (DNP) basis set that included all the occupied atomic orbitals plus a second set of valence atomic orbitals and polarized d-valence orbitals [63]; the correlation generalized gradient approximation (GGA) was applied, including BLYP setting [64]. When interacting molecules were initially separated by van der Waals distances, the DFT-D Grimme approximation was included [65]. All electrons were treated explicitly and the real space cutoff of 5 Å was imposed for numerical integration of the Hamiltonian matrix elements. The self-consistent field convergence criterion was set to the root mean square change in the electronic density to be less than 10^−6^ electron/Å^3^. The convergence criteria applied during geometry optimization were 2.72 × 10^−4^ eV for energy and 0.054 eV/Å for force.

### 3.3. Hydrodynamic Voltammetry at a Rotating Ring-Disk Electrode (RRDE)

Stock solutions of dexamethasone and colchicine (0.02 M) in anhydrous DMSO were used in trials. For the experiment, a solution of 0.1 M TBAB in anhydrous DMSO was bubbled for 5 min with a dry O_2_/N_2_ (35%/65%) gas mixture to establish the dissolved oxygen level in the electrochemical cell. The Au disk electrode was then rotated at 1000 rpm, while the disk was swept from 0.2 to 1.2 Volts and the ring was held constant at 0 Volts; the disk voltage sweep rate was set to 25 mV/s. The molecular oxygen reduction peak (O_2_ + e^−^ ---> O_2_^∙^) was observed at the disk electrode at 0.6 volts; the oxidation current (O_2_^∙^ ----> O_2_ + e^−^) was observed at the ring electrode. An initial blank was run on this solution and the ratio of the peak ring current to disk current was calculated as the “collection efficiency” in the absence of antioxidant. Next, an aliquot of the antioxidant is added, the solution is bubbled with the gas mixture for 5 min and the cyclovoltammogram is rerecorded. Aliquots (μL) for dexamethasone were 2, 2, 2, 4, 8, 8, 24 and 16; those for colchicine were 5, 15, 20, 25, 30, 35, 50 and 55. Again, the reduction and oxidation peaks were measured and the collection efficiency was calculated. Any decrease in the collection efficiency was due to the amount of superoxide removed by the antioxidant. A more detailed description of this technique can be found elsewhere [61].

## 4. Conclusions

Using molecular mechanics docking techniques, this study shows that four well-known anti-inflammatory drugs, which all have activity against COVID-19, also show inhibitory effects on caspase-1. Since the human enzyme is associated with triggering a potentially lethal cytokine storm, released by the immune system during COVID-19 infection, our results help to explain why these four inhibitory drugs are useful for the treatment of COVID-19 illness. The mechanism described for the caspase-1 active site indicates (1) that Cys285 thiolate formation (S–H cleavage) is due to proton capture by Ser339-O; (2) protonation of the O(carbonyl) drug after proton release from cationic His237, which activates the corresponding drug C(carbonyl); (3) an S(Cys285)-thiolate nucleophilic attack on the electron-deficient carbocation C(carbonyl), establishing a S–C(carbonyl) covalent bond and thus inhibiting caspase-1. We conclude that caspase-1 is an interesting target for developing other potential drugs active against COVID-19. It has also been shown that caspase-8 can induce the expression of pro-inflammatory cytokines and process pro-IL-1ß and IL-18 in the same way as caspase-1, resulting in the release of bioactive cytokines, through either pyroptosis or necroptosis [66]. Indeed, co-transfection of caspase-8 with pro-IL-1ß resulted in IL-1ß processing and secretion similar to that associated with caspase-1 [67]. Interestingly, a recent study on caspase-8 suggests, consistently, that caspase-1 might also assist towards the development of therapeutic strategies to treat COVID-19 [68].

The RRDE method applied to dexamethasone showed that it has no antioxidant scavenging activity for the superoxide radical, in agreement with DFT results. DFT results characterize colchicine as a scavenger of superoxide by capturing its radical electron and generating O_2_. More importantly, humidity in the DMSO solvent, or in the environment, suggests that the results of the RRDE experiment support the nucleophilic attack of superoxide to the protonated carbonyl of dexamethasone (seen after the initial addition of 5 μL) and colchicine. As for the anti-COVID-19 natural product celastrol [47], this suggests a parallel action regarding Cys285 thiolate action on the protonated carbonyl of dexamethasone and colchicine.

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
