# Peer review of "Mechanism of Caspase-1 Inhibition by Four Anti-inflammatory Drugs Used in COVID-19 Treatment"

_ijms, 2022, doi:10.3390/ijms23031849_

Round 1
Reviewer 1 Report
This is a resubmitted manuscript, which is clearly better than the previous version. Nonetheless, it still contains many unclarities that need to be improved before it can be considered for a publication:
- Please use the template provided by MDPI with line numbers.
- "In this study these three compounds together with one other agent known to be efficient against Covid-19" I think effective is more appropriate. Consider changing.
- "Several drugs previously used to modulate immune responses in many diseases have now been repurposed and used as promising new tools for reducing the inflammatory response to Covid-19 virus infection." This sentence needs citations. Consider adding these publications (PMID: 33283567, 33192602, etc.)
- Regarding "cytokine storm" in paragraph 1, please add a brief explanation about the definition of cytokine storm and cytokine release syndrome (CRS).
- "In this study, we suggest that inhibition of Caspase-1, an enzyme involved in triggering the release of pro-inflammatory cytokines, is relevant to the mechanism of the anti-Covid-19 therapy" The authors definitely need to explain beforehand, why caspase-1 was specifically targeted? is there any evidence of the involvement of caspase-1 in COVID-19 leading to this hypothesis? Why not other caspases? as said by the authors later on (supported by the paper by Li et al, 2020; REF [48]), caspase-8 was also important in COVID-19 pathogenesis. So, why caspase-1?
- "Figure 1 shows the potential inhibitors of Caspase-1 in this study." Up to this point, the reasons for selecting those 4 agents remain unclear. Why did the authors choose them over the others, in addition to their promising use for COVID-19? This is important to clarify because I can also point out other drugs such as lopinavir/ritonavir, remdesivir, molnupiravir etc that could be studied., but why those 4?
- Also, since the results concluded similar properties, I am curious if there is any COVID-19 drugs that doesn't work via the inhibition of caspase-1? The authors need to show one example with an opposite finding. Otherwise, this seems to good to be true.
- "In eight studies involving 926 Covid-19 patients, 520 patients had standard-of-care therapy while..." please add the word "patients".
- While the introduction about colchicine is adequate, the other 3 drugs are not well introduced. Please expand the introduction by adding relevant information about their effect in relation with caspase-1.
- Regarding the intro about corticosteroids, it also lacks the explanation about MP. Please add some information about the clinical use of dexamethasone and MP in COVID-19, for example from this review (PMID: 34321903).
- This sentence about dexamethasone "it is of interest to analyze whether the inhibition of Caspase-1 is implicated in these cases" is contradictory with this one "Figure 1 shows the potential inhibitors of Caspase-1 in this study." Please clarify whether the caspase-1 inhibition by steroids has been reported before, or it is only based on the authors' speculation.
- It is also important to elaborate the authors' strategy to answer that particular question with their study. How could a computational docking study confirm that caspase-1 is involved in the corticosteroids-mediated effect on COVID-19?
- I am curious about the result of other steroids, for example, prednisone or budesonide? Do they also have similar caspase-1 inhibitory activity? consider adding the simulations.
- As I said earlier, the introduction about raloxifene is not enough. Please expand the discussion, including the known effect on caspase activity and inflammation.
- "In summary, we studied two drugs used against Covid-19, dexamethasone and methylprednisolone, plus colchicine and raloxifene, two other medicinal compounds showing anti-Covid-19 activity using computational methods. We also measured the antioxidant activity of dexamethasone and colchicine towards the damaging superoxide radical using the RRDE cyclovoltammetry method, recently developed in our laboratory." Why didn't I see caspase-1 anymore in here? Please rephrase this paragraph reflecting the aim of the study as projected in the title and abstract.
- Please also separate the result and discussion sections. They serve different purposes and mixing them is very distracting because it is hard to differentiate between the findings of this study and the subsequent analysis that may involve data from other publications.
- Please also check the order of the figures. They are messed up, some Figure S... are in the middle of the main figures. I strongly suggest to select max. 8 figures to be displayed in the main manuscript and the rest can go to a separate supplementary file.
- The conclusion needs to be rewritten. Please only include 4-5 sentences about the highlights / core findings of the study.
Reviewer 2 Report
The manuscript was significantly improved and could be accepted in present form
Author Response
We thank that Reviewer 2 has accepted our previous revision.
Round 2
Reviewer 1 Report
Thanks for the responses. I have no further remarks.
Author Response
Thanks for accepting all our responses
This manuscript is a resubmission of an earlier submission. The following is a list of the peer review reports and author responses from that submission.
Round 1
Reviewer 1 Report
The investigations concerning the potential mechanism of caspase-1 inhibition of five drugs is interesting, but some mistakes should be corrected before final decision and publishing.
First of all, many grammar and spelling mistakes occur in the manuscript. The English language have to be corrected.
In the introduction the authors did not mentioned why did they investigate methylprednisolone. It is only said in line 160. It should be corrected. The introduction should clarify the selection of all drugs.
Besides it, introduction contains sufficient information, but its compisition and language must be improved. For example: sentences in lines 44-48 concerning performed investigations should take place it the end, not at the beginning of introduction.
Scheme 1. In fact, it is not a scheme - it does not present any synthetic pathway. It is a figure. Moreover the form of structures presented in this "scheme" have to be corrected. They have different size ect. The fluorine atom in dexamethasone is bigger and written in different font. Generally, this figure is too big and takes a lot of place in manuscript.
Line 100 and 231- I do not agree that doxycycline is an anti-inflammatory drug. It is a tetracycline-class antibiotic which may exert anti-inflammatory effect. This serious mistake have to be corrected.
Firures 2 - 32. They are too big, they have different sizes and are deployed a little bit chaotic and uneven. Their size and positioning should be unified. Maybe some of them can be moved to supplementary data?
Why colchicine docking studies are described after dexamethasone and methylprednisole. For colchicine caspase-1 inhibition activity have been proved. So may be colchice should be described as a pattern and other drugs sholud be compared to it?
Reviewer 2 Report
The manuscript is well-written and interesting. However, I don't think this study is appropriate for publication in EJMS since there are no molecular studies. I very recommend choosing another journal.
Reviewer 3 Report
In this computational study, Dr. Caruso and colleagues investigated the mechanisms of caspase-1 inhibition in 5 drugs (colchicine, dexamethasone, methylprednisolone, doxycycline and raloxifene) that were shown by previous studies to be beneficial against COVID-19. Overall, the study can be insighful for further development of COVID-19 medications. However, several issues are present and need to be addressed by the authors:
- Grammatically, the English is okay but the writing style is not good. The sentences are not concise enough and therefore, it is hard to understand the main message of this manuscript. I think this manuscript can be significantly shortened if it is written appropriately. I would suggest the authors to consult with a scientific writer or to use the professional editing service to improve the quality of this paper.
- The abstract certainly needs to be rewritten. Too many mistakes in the writing that I cannot describe individually and those impair the readability of this manuscript. For example, we never say "the corticosteroid anti-inflammatory drugs dexamethasone". It is grammatically okay but the writing style is not good.
- Contentwise, from 5 drugs that were analyzed in this study, not all of them received the emergency authorization from the US FDA. In fact, only corticosteroids received it, which indicates the high likelihood to be beneficial in COVID-19. Therefore, it is not relevant to link COVID-19 with these drugs (except steroids) and the authors need to change the storyline and the background of the study if they want to keep these 5 drugs.
- There is no such things called "scheme 1" in a research article. It should be Figure 1.
- The proportion of the description about colchicine is disproportional compared to the other 4 drugs. This has to be improved. For once, I thought I was reading a narrative review. Please be more concise and to the point.
- Although there is no figure limitations in MDPI journals, I would suggest to keep max. 8 Figures in the main manuscript and the rest can go to the supplement. This is important to maintain the readability of the paper.
In general, please follow the common guide of scientific writing and please keep everything clear and concise.
Round 2
Reviewer 1 Report
The Authors have corrected most of previous mistakes. English Language is corrected, as well as introduction and overall presentation. Nevertheless the manuscript is still overloaded with figures and it results in low quality of presentation.
If the Editor accept such form of manuscript it may be published.
Reviewer 2 Report
The manuscript was improved and could be accepted.
Reviewer 3 Report
Thank you for at least trying to address my concerns. However, my comments remain and the manuscript is barely improved. I will try to elaborate again some and add some more suggestions for future submission.
- I strongly suggest the authors to involve one or more COVID-19 clinical/experimental experts as co-author(s) because this manuscript contains many fundamental flaws that I will exemplify below:
- Please do not make any false claim that raloxifene and doxycycline are effective for COVID-19. They are not yet confirmed and not recommended by FDA to be used in COVID-19. Many publications also objected their use in the disease.
- Also please distinguish between effective and efficient. They are not interchangeable.
- The authors can check this preprint to see what immunomodulatory drugs are being investigated clinically. (https://www.preprints.org/manuscript/202104.0022/v1)
- In the introduction, the reason why the authors chose caspase-1 over other caspases and inflammatory cytokines remains unclear. This has to be improved. They cannot suddenly say that they want to investigate the role of caspase-1 in paragraph 2 of the introduction without explaining what is it for in COVID-19 and what do we know about its significance in the pathogenesis of the disease.
- Speculation is okay at a certain level but basic understanding of the disease is a must. The authors said that the inhibition of caspase-1 could potentially ameliorate patient healing. Where is this hypothesis from? Certainly computational biochemistry and docking simulations are not adequate to confirm this claim.
- The writing style has to be improved. The coherence is lacking and everything is poorly organized. Please check how other already accepted papers in high impact journals were written. The manuscript has to be succint, clear, coherent and valid.
- Please also check how to report a figure in a paper. The authors need legends to explain what is on the figure even though it has been explained in the text.
- I am not sure if these results are adequate to confirm the roles of caspase-1 inhibition. Functional studies and experiments are needed to justify the claim and these results could be a companion to those experiments but not as a stand alone data. The authors need to convince me and the future readers on the relevance and significance of the study.